# Determining Work-Rest Schedules for Visual Tasks That Use Optical Head-Mounted Displays Based on Visual Fatigue and Visually Induced Motion Sickness Recovery

**DOI:** 10.3390/ijerph20031880

**Published:** 2023-01-19

**Authors:** Chih-Yu Hsiao, Chia-Chen Kuo, Yi-An Liou, Mao-Jiun Wang

**Affiliations:** 1Department of Industrial Engineering and Engineering Management, National Tsing Hua University, Hsinchu City 30013, Taiwan; 2Department of Industrial Engineering and Management, National Chin-Yi University of Technology, Taichung City 41170, Taiwan; 3Department of Industrial Engineering and Enterprise Information, Tunghai University, Taichung City 40704, Taiwan

**Keywords:** visual task, electroencephalography, critical flicker fusion frequency, Simulator Sickness Questionnaire

## Abstract

This study aimed to determine work-rest schedules for visual tasks of different lengths by evaluating visual fatigue and visually induced motion sickness (VIMS) using an optical head-mounted display (OHMD). Thirty participants were recruited to perform 15 and 30 min visual tasks using an OHMD. After completing each visual task, participants executed six levels of rest time. Critical flicker fusion frequency (CFF) values, relative electroencephalography indices, and Simulator Sickness Questionnaire (SSQ) scores were collected and analyzed. Results indicated that after completing the 15 and 30 min visual tasks, participants experienced visual fatigue and VIMS. There was no significant difference between baseline CFF values, four electroencephalography relative power index values, and SSQ scores when participants completed a 15 min visual task followed by a 20 min rest and a 30 min visual task followed by a 30 min rest. Based on our results, a 20 min rest for visual fatigue and VIMS recovery after a 15 min visual task on an OHMD and a 25 min rest for visual fatigue and VIMS recovery after a 30 min visual task on an OHMD are recommended. This study suggests a work-rest schedule for OHMDs that can be used as a reference for OHMD user guidelines to reduce visual fatigue and visually induced motion sickness.

## 1. Introduction

With the rapid development of display technologies, augmented reality (AR) has emerged as a next-generation display technology for human–computer interactions. Optical head-mounted displays (OHMDs) are becoming increasingly popular in daily activities. Most OHMDs use microdisplays, eyepieces, and optical combiners to present AR images on a screen [1]. The eyepiece magnifies a digital image from the microdisplay and forms an AR image that appears at a distance from the eye. The combiner superimposes the AR image of the OHMD on a real-world scene [1,2]. There are two common AR image presentation types in OHMDs: the screen-fixed type, where the AR image is displayed at a fixed position on the OHMD display screen, and the world-fixed type, where the AR image is displayed at a fixed position in a real-world environment [3,4]. Most OHMDs present AR images at a fixed focal distance [5]. The mismatch between the focal distance of a real-world scene and AR images results in the switching of accommodation and attention between the real-world scene and the AR images. Frequent switching of accommodation between different focal depths (e.g., real-world scenes and AR images) can result in vergence–accommodation conflicts [6] and visually induced motion sickness (VIMS) [7].

Vergence–accommodation conflict is a common problem with virtual reality head-mounted displays (VR HMDs) and AR HMDs [8], which are usually designed to display digital images at a single focal depth [9]. When viewing digital images through an HMD, the user may feel that the positional depth of the digital image does not match that of the physical object in the real environment. Thus, the user must adapt to provide a clear view of digital images and physical objects placed at different depths in their environment [9]. However, the user’s brain will have to adapt unnaturally to the conflict and increase the fusion time of binocular images while reducing the fusion accuracy [10]. Gabbard et al. [5] demonstrated that vergence–accommodation conflict affects visual fatigue and user performance. Park et al. [11] indicated that visual fatigue is a performance decrement of the human visual system caused by the extraocular and ciliary muscle over-exertion.

Critical flicker fusion frequency (CFF or CFFF) is a commonly used method to measure visual fatigue, and it is defined as the lowest frequency at which intermittent light stimulation appears to be steady to the observer [12]. Chi and Lin [13] demonstrated that an increase in the duration of visual tasks is associated with a decrease in the CFF value. Moreover, Wu [14] indicated that when the CFF value was measured before and after a visual task, if the CFF value after a visual task decreased by at least 1 Hz compared to the CFF value measured before the task, visual fatigue was considered. In addition to CFF values, Hsu and Wang [15] suggested that electroencephalography (EEG) recorded from the occipital lobe was suitable for measuring visual fatigue. Furthermore, Fisch [16] revealed that EEG rhythms can be divided into four bands using fast Fourier transform (FFT), namely alpha (α, 8–13 Hz), beta (β, 13–30 Hz), theta (θ, 4–7 Hz), and delta (δ, 0.5–4 Hz). Zou et al. [17] measured the EEG signals and CFF values of 3D monitor users, and their results indicated that as the viewing time increased, θ activity stabilized, α activity increased significantly, and there was a substantial reduction in β activity. The correlation between the EEG signals and CFF values suggests that α is the most promising index for detecting stereoscopic fatigue. There were significant differences in EEG signals among the participants of these studies. However, the EEG relative power indices can effectively reduce the influence of individual differences among participants [15]. The EEG relative power indices were computed by dividing the power in each band by the sum of the powers from all bands [15]. Several studies have indicated that basic and ratio EEG relative power indices can also be used to measure visual fatigue levels [15,18,19]. Moreover, Ramadan et al. [20] suggested that when viewing 3D displays, the EEG relative power of the β band decreased and the α/β ratio increased throughout the viewing cycle.

VIMS is a medical condition in which nausea, disorientation, and oculomotor symptoms occur when the vestibular system senses acceleration that is not perceived (or perceived differently) by the visual system [7]. Extensive studies have been conducted on the VIMS symptoms caused by simulators or VR [21]. The Simulator Sickness Questionnaire (SSQ) is a common instrument for evaluating VIMS; it consists of 16 symptom lists that reflect the current status of VIMS. Each symptom is rated on a four-point Likert scale [22]. Importantly, the SSQ provides an overall severity score as well as subscales for nausea, oculomotor symptoms, and disorientation. According to Saredakis et al. [23], disorientation is the most serious VIMS symptom after using VR HMD, followed by oculomotor nerve symptoms and nausea symptoms. Kaufeld et al. [7] discussed the severity of VIMS symptoms after using an AR HMD, and their results indicated that oculomotor symptoms were the most serious, followed by disorientation and nausea.

Many studies have been conducted on computers or handheld smart devices to prevent and alleviate the visual fatigue caused by viewing digital screen devices. Reddy et al. [24] developed common strategies to prevent and relieve visual fatigue when using digital screen devices, including scheduling rest time, looking at distant objects away from the screen, massaging the eyes, and using eye drops. Regarding relieving VIMS, Keshavarz et al. [25] suggested that pleasant odors were more effective in relieving VIMS than unpleasant or no odors. Furthermore, Keshavarz and Hecht [26] demonstrated that playing relaxing music can reduce the severity of VIMS. To examine the effects of taking a break, Moss and Muth [27] required participants to wear a VR HMD for 20 min and then rest to investigate rest schedules when using VR HMDs. After a break of 5 min, the participants’ SSQ scores were significantly higher than their scores at baseline. In contrast, the SSQ scores decreased significantly after a break of 10 min, almost returning to the baseline.

To examine the effects of taking a break, Shieh and Chen [28] reported that the work-rest schedule affected the view distance during VDT work. Users with a 25 min work period and a 5 min rest period had greater viewing distances than users with a 50 min work period and 10 min rest period. Balci and Aghazadeh [29] indicated that a 30 min work period and a 5 min rest period followed by a 15 min work period and micro-rest showed the least eyestrain and blurred vision when users performed data entry on VDT. Hayashi et al. [30] proposed that a work-rest schedule of approximately 60 min of work and a 20 min nap for rest when using a visual display terminal could reduce mental fatigue and help in maintaining a high subjective alertness and performance level. Moreover, Wu et al. [31] recommend that smartphones with an active-matrix organic light-emitting diode (AMOLED) display should not be used to read for more than 64 min. 

User experiences when viewing AR information on OHMDs are very different from their experiences when using computers, handheld smart devices, or VR HMDs. In contrast to the virtual environment constructed by VR HMDs, when using OHMDs, users have the visual experience of watching AR objects and the real environment simultaneously. In addition, when viewing or controlling computers or handheld smart devices, users can easily change the viewing distance between themselves and the display, whereas traditional OHMD AR images appear at a fixed distance from the eyes [32]. Therefore, the work-rest schedule previously proposed for computers, handheld smart devices, and VR HMDs may not be applicable to OHMDs.

With the increasing interest in OHMDs, they are being widely used in various fields. Performing visual tasks on OHMDs for any duration induces visual fatigue and VIMS. Gao et al. [33] indicated that the CFF value decreases as visual task duration increases with optical see-through head-mounted displays (OST-HMDs). Kaufeld et al. [7] detected VIMS symptoms while participants performed both static and dynamic AR tasks while participants wore the OHMD. How to minimize visual fatigue and VIMS involved in using OHMD remains an important challenge. Therefore, it is necessary to determine an appropriate work-rest schedule to alleviate visual fatigue and VIMS symptoms that may be caused by the use of OHMDs. Thus, this study aimed to determine the appropriate work-rest schedules for OHMD visual tasks of different lengths by evaluating visual fatigue and VIMS.

## 2. Materials and Methods

### 2.1. Participants

This study included 30 healthy male participants. Their ages ranged from 21 to 35 years (mean = 28.41, standard deviation = 2.74 years). All the participants were required to have normal or corrected-to-normal vision. The participants were requested not to use digital devices or watch television (TV) 1 h before data collection to avoid visual discomfort and fatigue.

### 2.2. Apparatus

Visual acuity was measured using an OPTEC 2000P Vision Tester (STEREO OPTICAL Inc., Chicago, IL, USA). Laboratory illumination was measured using an illuminometer (TES-1337B, TES Electrical Electronic Co., Taiwan). Epson Moverio BT-200 (Epson Inc., Los Alamitos, CA, USA) AR glasses were the OHMD used in this study. The display projector had a 960 × 540 red–green–blue resolution and provided a 23° field of view [34]. All participants were able to adjust their nose pads for comfort. A CFF tester (instrument model No. 501c, Takei Kiki Kogyo Co., Niigata, Japan) was used to evaluate the participants’ visual fatigue. EEG signals were acquired using a portable amplifier and EEG data recording device that incorporated a 24-bit A/D converter (NeXus-10, MindMedia, Herten, The Netherlands) with BioTrace + software (MindMedia, Herten, the Netherlands).

### 2.3. Experimental Design

A full factorial design was employed with two factors: length of the visual task time and length of rest time. Participants were asked to watch an experimental movie on an OHMD. The visual task involved two levels of visual task duration and six levels of rest duration. The experimental movie content in this study included a documentary, comedy, action, romance, science fiction, or drama, chosen by each participant. All participants chose a drama movie as the experiment movie. The sequence of the visual task duration for each participant was randomly arranged. Prior to each visual task, critical flicker fusion frequency (CFF) values, electroencephalogram (EEG) signals, and Simulator Sickness Questionnaire (SSQ) scores were recorded as baseline data. After completing each visual task, the CFF values, EEG signals, and SSQ scores were recorded again. Upon completion of each visual task, a 30 min rest was provided with CFF values, EEG signals, and SSQ scores recorded every 5 min.

The video was projected onto the front visual field of the participants, and the off-axis viewing angle was not investigated in this study. In this study, the OHMD luminance was 500 cd/m^2^ and it remained constant throughout the experiment in all tests. The laboratory temperature was maintained at 25 °C, and illumination in the laboratory environment was maintained at 500–600 lx. A white projection screen was used to provide the standard experimental background. Participants wore an OHMD and faced the projection screen, which was placed 2 m in front of them, as shown in Figure 1.

#### 2.3.1. Independent Variables

Previous studies have proposed specific work-rest schedules with VDT or handheld smart devices or VR HMDs to prevent and alleviate visual fatigue and VIMS caused by viewing digital screen devices. However, these recommendations do not account for OHMD effects on visual fatigue and VIMS. It remains necessary to examine appropriate OHMD visual task lengths and rest times considering how to prevent and alleviate visual fatigue and VIMS. Therefore, visual tasks and rest durations were independent variables. Two levels of visual task duration (15 and 30 min) and six levels of rest duration (5, 10, 15, 20, 25, and 30 min) were specified in this study, following the modified visual task periods and rest periods in the study by Moss and Muth [27], Shieh and Chen [28], and Balci and Aghazadeh [29].

#### 2.3.2. Dependent Variables

Dependent variables included the CFF values, EEG relative power indices, and SSQ scores. In this study, the CFF value and EEG relative power indices were used to measure the severity of visual fatigue and recovery from visual fatigue. CFF values were collected for both eyes, and the mean value of three ascending trials (flicker-to-fusion trial) and three descending trials (fusion-to-flicker trial) was used as the CFF value [13].

After completing the visual tasks and during rest, EEG signals were measured. EEG Ag/AgCl electrodes were attached to specific locations on each participant’s scalp. The exploring electrodes were located over the left occipital lobe (O1) and right occipital lobe (O2); the reference electrodes were located on the prominent bone just behind the left ear (A1) and that just behind the right ear (A2). The electrode locations were based on the international 10–20 system [16]. EEG signals were monitored at 1024 Hz, and raw data were processed using the BioTrace+ software with an IIR bandpass filter. Prior to data processing, a 60 Hz notch filter was applied to remove environmental artifacts [15]. With respect to the EEG signals, relative power indices, α, β, θ, θ/α, and β/α, were used as dependent variables for subsequent analyses.

The SSQ scores, comprising 16 symptoms that reflect the severity of visually induced motion sickness (VIMS), were used in this study to measure VIMS severity and recovery. Participants rated each SSQ item on a 4-point Likert scale (0 = not at all, 1 = slight, 2 = moderate, 3 = severe) [22]. Kennedy et al. [22] indicated that the total severity score is obtained by adding each SSQ item scale multiplied by the weighted score of nausea, oculomotor symptoms, and disorientation, as presented in Table 1.

### 2.4. Experimental Procedure

The experimental environment was standardized prior to data collection. The SSQ scores were checked to ensure that each participant had no VIMS symptoms at the outset, indicating a score of 0 in the SSQ total severity score. The purpose and procedures of the experiment were explained to the participants in detail. Participants were requested to complete informed consent forms. Subsequently, each participant’s chair and nose pad were adjusted for comfort, and CFF values, EEG signals, and SSQ scores were then measured as baseline values.

Visual task conditions and movie content were arranged randomly. At the beginning of each visual task, a movie was presented on the OHMD and participants were asked to focus their attention on watching the movie. Upon completion of the visual task, a 30 min rest was provided, and CFF values, EEG signals, and SSQ scores were recorded every 5 min. The experimental procedure sequence is shown in Figure 2. Each participant completed two visual tasks on two separate days.

### 2.5. Statistical Analysis

Statistical analyses using analysis of variance (ANOVA) and a Wilcoxon signed-rank test were conducted to evaluate how the length of the visual task time and length of the rest period affect the CFF value, EEG relative power indices, and SSQ scores. The Kolmogorov-Smirnov test was performed to examine the normality of the data distribution if the normality assumption was confirmed. If the normality assumption was violated, the Wilcoxon signed-rank test was used for corresponding variables. Duncan’s multiple range tests were conducted as post hoc tests. In addition, the paired *t*-test and Wilcoxon signed-rank *t*-test were used to examine the EEG electrode locations’ (O1 and O2) effects on the five EEG relative power indices. Spearman correlation analysis was used to evaluate the relationship between the CFF value and relative EEG power indices. The statistical significance level was set at *p* = 0.05. All statistical analyses were performed with SPSS Statistics version 22.0 (IBM Corp., Armonk, NY, USA) software. Line graphs were made with Excel version 23.0 (Microsoft Corp., Redmond, WA, USA).

## 3. Results

The results of the Kolmogorov–Smirnov test indicated that the CFF values, EEG relative power indices, and SSQ scores violated the normality assumption. The post-experiment CFF values were significantly lower than those at baseline in the two visual tasks (15 min: Z (240) = 4.712, *p* < 0.001; 30 min: Z (240) = 4.551, *p* < 0.001).

The results of CFF values are presented in Table 2. The greatest CFF value occurred at baseline in both the 15 min and 30 min visual tasks. In addition, the lowest CFF value occurred in the post-experiment period in both the 15 min and 30 min visual tasks. The results showed that the participants had higher severity of visual fatigue after executing a 30 min visual task than after executing a 15 min visual task. The CFF value increased with the rest time. After the 15 min visual task and 10 min rest, there were no significant differences in CFF values at baseline and at the end of the 10 min rest time (Z (240) = 1.621, *p* = 0.110). Additionally, after the 30 min visual task and 15 min rest, there were no significant differences in CFF values at baseline and at the end of the 15 min rest period (Z (240) = 1.661, *p* = 0.100).

For EEG relative power indices, the Wilcoxon signed-rank test was conducted to examine the effects of EEG electrode location (O1 and O2) on the five EEG relative power indices, as shown in Table 3. The result shows the five EEG relative power indices obtained from the left and right occipital regions (15 min visual task: relative α power index Z (240) = −0.23, *p* = 0.83, relative β power index Z (240) = −0.21, *p* = 0.63, relative θ power index Z (240) = −0.49, *p* = 0.63, relative β/α power index Z (240) = −0.44, *p* = 0.66, and relative θ/α power index Z (240) = −0.03, *p* = 0.98; 30 min visual task: relative α power index Z (240) = −0.66, *p* = 0.51, relative β power index Z (240) = −0.44, *p* = 0.66, relative θ power index Z (240) = −0.74, *p* = 0.46, relative β/α power index Z (240) = −0.95, *p* = 0.34, and relative θ/α power index Z (240) = −0.11, *p* = 0.91). Hence, analyzing the EEG relative power index data from one occipital lobe electrode site was sufficient.

The 15 min visual task had significant effects on four EEG power indices (relative α index (Z (240) = 2.951, *p* = 0.003), relative β index (Z (240) = 2.401, *p* = 0.017), relative θ index (Z (240) = 3.671, *p* < 0.001), and relative β/α index (Z (240) = 3.262, *p* = 0.001)), but no significant effects on θ/α (relative θ/α index (Z (240) = 1.242, *p* = 0.213)), as shown in Figure 3.

After a 15 min visual task, the relative α index value significantly increased (Z (30) = −2.915, *p* = 0.003); the relative β index value had significantly decreased (Z (30) = −2.851, *p* = 0.003); the relative θ index value had significantly increased (Z (30) = −2.405, *p* = 0.017); the relative β/α index value had significantly decreased (Z (30) = −3.256, *p* = 0.001). This shows that after completing the visual task, participants had visual fatigue. The relative β and β/α indices decreased with increasing rest times. After a 10 min rest, there were no significant differences in relative β index value at baseline and at the end of the 10 min rest period (Z (30) = −1.841, *p* = 0.066); after a 20 min rest, there were no significant differences in relative β/α index value at baseline and at the end of the 20 min rest period (Z (30) = −1.244, *p* = 0.213). The relative α and θ indices increased with increasing rest times. After a 20 min rest, there were no significant differences in relative α index value at baseline and at the end of the 20 min rest period (Z (30) = −0.782, *p* = 0.482), and no significant differences in relative θ index value at baseline and at the end of the 20 min rest period (Z (30) = −1.676, *p* = 0.094). Additionally, after the 15 min visual task and 20 min rest, there were no significant differences in the four EEG power indices at baseline and at the end of the 20 min rest.

The 30 min visual task had significant effects on four EEG power indices (relative α index (Z (240) = 4.021, *p* < 0.001), relative β index (Z (240) = 2.792, *p* = 0.005), relative θ index (Z (240) = 3.202, *p* = 0.001), and relative β/α index (Z (240) = 4.062, *p* < 0.001)), but had no significant effect on θ/α (relative θ/α index (Z (240) = 1.201, *p* = 0.229)), as shown in Figure 4.

After a 30 min visual task, the relative α index value significantly increased (Z (30) = −2.478, *p* = 0.013); the relative β index value had significantly decreased (Z (30) = −2.854, *p* = 0.004); the relative θ index value had significantly increased (Z (30) = −1.944, *p* = 0.041); the relative β/α index value had significantly decreased (Z (30) = −2.232, *p* = 0.026). This shows that after completing the visual task, participants had visual fatigue. The relative β and β/α indices decreased with increasing rest times. After a 30 min rest, there were no significant differences in relative β index value at baseline and at the end of the 30 min rest period (Z (30) = −0.031, *p* = 0.975); and no significant differences in relative β/α index value at baseline and at the end of the 25 min rest period (Z (30) = −0.216, *p* = 0.829). The relative α and θ indices increased with increasing rest times. After a 25 min rest, there were no significant differences in relative α index value at baseline and at the end of the 25 min rest period (Z (30) = −0.031, *p* = 0.926); after a 30 min rest, there were no significant differences in relative θ index value at baseline and at the end of the 30 min rest period (Z (30) = −0.041, *p* = 0.494). Additionally, after the 15 min visual task and 30 min rest, there were no significant differences in the four EEG power indices at baseline and at the end of the 30 min rest.

The 15 and 30 min visual tasks had significant effects on nausea, oculomotor function, disorientation, and total scores (nausea scores in 15 min (Z (240) = 2.952, *p* = 0.003) and nausea scores in 30 min (Z (240) = 4.011, *p* < 0.001); oculomotor scores in 15 min (Z (240) = 4.744, *p* < 0.001) and oculomotor scores in 30 min (Z (240) = 4.713, *p* < 0.001); disorientation scores in 15 min (Z (240) = 3.854, *p* < 0.001) and disorientation scores in 30 min (Z (240) = 4.414, *p* < 0.001); total severity scores in 15 min (Z (240) = 4.721, *p* < 0.001) and total severity scores in 30 min (Z (240) = 4.715, *p* < 0.001)), as presented in Table 4 and Table 5, respectively. The 30 min visual task induced higher nausea, oculomotor function, disorientation, and total scores than the 15 min visual task. For SSQ scores taken after the 15 min visual task followed by a 5 min rest, the disorientation (Z (30) = 1.254, *p* = 0.315) was not significantly different from baseline. In addition, after a 15 min rest, nausea (Z (30) = 3.124, *p* = 0.101), oculomotor (Z (30) = 2.245, *p* = 0.211), and total scores (Z (30) = 4.215, *p* = 0.345) were not significantly different from baseline. After the 30 min visual task followed by a 15 min rest, the disorientation score (Z (30) = 2.245, *p* = 0.445) was not significantly different from baseline. In addition, after a 20 min rest, nausea (Z (30) = 2.224, *p* = 0.211), oculomotor (Z (30) = 3.12, *p* = 0.351), and total scores (Z (30) = 3.152, *p* = 0.245) were not significantly different from baseline.

Spearman correlation was conducted to investigate the relationship between the four relative EEG power indices and the CFF value for the visual task duration and rest duration, as presented in Table 6. For the 15 and 30 min visual task durations, the CFF value showed significant negative correlations for visual task duration and rest duration. The relative α, β, θ, and β/α power indices showed no significant correlation for either visual task duration. In addition, the CFF value, relative α, β, θ, and β/α power indices showed no significant correlation for either rest durations for the 15 and 30 min visual tasks.

## 4. Discussion

This study aimed to determine work-rest schedules for visual tasks of different lengths by evaluating visual fatigue and VIMS while using an OHMD. The results indicated that decreased CFF values were observed after participants completed 15 and 30 min visual tasks at 1.48 Hz and 2.02 Hz, respectively. Wu et al. [14] identified visual fatigue as a decline in CFF values by at least 1 Hz after the visual task. Thus, the decreased CFF values in this study indicated that visual fatigue occurred after the 15 and 30 min OHMD visual tasks.

Table 3 shows that there was no significant difference between EEG relative power index data from O1 and O2. The results were similar to those of Hsu and Wang [15], who indicated that there is no significant difference in EEG power indices collected between the left and right occipital lobes. Analyzing the EEG power indices from one occipital lobe electrode site was deemed valid and sufficient. Therefore, this study analyzed EEG relative power index data from one occipital lobe electrode site.

Figure 3 and Figure 4 show that the OHMD visual task had a significant effect on the relative α, β, θ, and β/α indices, but no significant effect on the relative θ/α index. This is consistent with a previous study that showed that sustained visual tasks affect EEG relative power indices [15]. Hsu and Wang [15] reported that sustained visual tasks increase the visual load and influence neurophysiological function in the brain. This could explain why EEG relative power indices changed after each visual task was completed. Furthermore, the relative α index increased and the relative β and β/α indices decreased after each visual task was completed. The results of this study were consistent with those reported by Zou et al. [17] and Hsu and Wang [15]. Zou et al. [17] indicated that α activity increases and β activity decreases after watching 3DTV. Hsu and Wang [15] indicated that relative α, β, and β/α indices are effective in evaluating visual fatigue when playing TV games. A decrease in relative β and β/α indices and an increase in the relative α index are associated with visual fatigue.

In this study, the participants’ relative θ index significantly increased after executing the OHMD visual tasks. The relative θ index trend contradicted the results reported by Hsu and Wang [15]. One explanation for this discrepancy is the difference in task type. Hsu and Wang [15] evaluated the effects of different lengths of TV gameplaying time on visual fatigue. This study evaluated visual fatigue while watching a movie on an OHMD. In contrast to TV video games, the participants in this study may have focused their attention on movie pictures and content in the experiment. Boksem et al. [35] reported that the θ power index provided valid results for detecting fatigue in static tasks. Lai and Craig [36] indicated that EEG θ waves might be produced by static and memory tasks, and that θ wavebands might increase during drowsiness or sleep. Hence, it is reasonable to infer that the level of drowsiness increases with increasing visual task time.

For visual fatigue recovery, after completing a 15 min visual task followed by 10 min of rest and a 30 min visual task followed by 15 min of rest, the participants’ CFF values were not significantly different from baseline. After completing a 15 min visual task followed by 20 min of rest and a 30 min visual task followed by 25 min of rest, the four EEG relative power indices showed no significant difference between the baselines. Hsu and Wang [15] reported that CFF variation reflects decreased retinal or optic nerve activity. On the other hand, visual fatigue-related EEG power indices reflect the electrical activity between the retina and visual cortex. In addition to this electrical activity between the retina and visual cortex, Iwasaki and Akiya [37] reported that muscular workload, mental fatigue, and subjective feelings might also induce visual fatigue-related EEG power indices. Cajochen et al. [38] also indicated that changes in EEG power indices appeared when subjective fatigue was manifested. This may explain why the recovery time for the CFF value was shorter than that for the EEG relative power index value. This implies that some physiological signals and subjective feelings influence visual fatigue perception, and that participants require more rest time to recover.

VIMS severity level increased with increasing visual task time, as shown in Table 4 and Table 5. This result was consistent with that of a previous study that showed that visual task time length affects VIMS severity levels [27]. Table 4 and Table 5 also show that the post-experiment SSQ scores were significantly higher than baseline SSQ scores. This result is consistent with that of Drexler [39], in which the largest relative sickness contribution in VIMS was oculomotor symptoms, followed by disorientation and nausea. OHMDs use sophisticated visual display technology to present augmented reality (AR) objects for several applications. Hence, participants will show increased oculomotor symptom severity during or after exposure to certain strong dynamic visually perceived motions [40]. It should be noted that the standard deviation of the SSQ scores was large. Kennedy et al. [22] reported that participants rated the severity of their symptoms as none (0), mild (1), moderate (2), or severe (3). These ratings were then multiplied by the weightings such that a few ratings reached a score of 10 or above. This may explain why the SSQ scores in this study had large standard deviations. The results indicated that VIMS severity decreased with increasing rest time. After the 15 min visual task and 15 min rest and 30 min visual task and 25 min rest, participants’ VIMS severity levels were not significantly different from baseline levels.

Apart from the SSQ for VIMS severity level measurement, Liu et al. [41] used EEG power indices for VIMS evaluation when participants used a VR-based vehicle-driving simulator, and indicated that the mean gravity frequency of the θ wave in the frontal lobe areas (FP1 and FP2), α waves in the temporal lobe areas (TP9 and TP10), α waves in the frontal lobe area (FP2), and β waves in the frontal lobe area (FP1) decreased significantly in the VIMS state. The current study did not investigate the correlation between relative EEG power indices and VIMS when the participants used OHMDs. This can be implemented in future studies to determine how VIMS affects the relative EEG power indices while using OHMD, as well as to understand how the types of relative EEG power indices can be effectively used to evaluate VIMS when using OHMDs.

The results of this study showed that after completing a 15 min visual task followed by a 20 min rest and a 30 min visual task followed by a 25 min rest, the participants’ CFF values, four EEG relative power index values, and SSQ scores showed no significant difference from baseline. Thus, this study suggests a 20 min rest for visual fatigue and VIMS recovery after a 15 min visual task on an OHMD and a 25 min rest for visual fatigue and VIMS recovery after a 30 min visual task. Yoshimura and Tomoda [42] recommended that the work-rest schedule for VDT work is 50 min of VDT working time and 15 min of rest. Balci and Aghazadeh [29] suggest 10–20 min of rest for fatigue recovery after 50–60 min of VDT work. The OHMD working time suggested in this study was shorter than the VDT working time suggestions in previous studies. Kramida [8] reported that vergence–accommodation conflict is a common problem with AR HMD, and Kaufeld et al. [7] indicated that when AR OHMDs are used, severe VIMS symptoms may increase. The differences in visual demand between the VDT and OHMD tasks resulted in the differences in the work-rest schedule between the two types of tasks.

Table 6 shows the correlation analysis results. The CFF value is sensitive in the 15 and 30 min visual task and rest durations. Hsu and Wang [15] reported that the CFF value’s decreased deterioration reflects the retinal function and optic nerve activity. However, Hsu and Wang [15] showed that α, β, θ/α, β/α, (α + θ)/β, and CFF are sensitive in short-term tasks, which differs from the results of this study. One possible reason is that the visual task times for this study were 15 min and 30 min, while the short-term task time by Hsu and Wang [15] was 60 min. In the future, the OHMD visual task time may be increased to conduct the most suitable evaluation of visual fatigue and visual fatigue recovery indicators.

When using an OHMD, the visual fatigue level and severe VIMS symptoms may increase the risk of injury at the workplace or during daily activities. Hence, this study provides a work-rest schedule for OHMD use: a 15 min visual task and 20 min rest, and 30 min visual task and 30 min rest, to alleviate the users’ visual fatigue and VIMS severity levels, and to enhance the safety and health in the workplace. The current study proposed a work-rest schedule of a 15 min visual task period followed by a 20 min rest break and a 30 min visual task period followed by a 25 min rest period, as the participants’ severity of visual fatigue and VIMS did not significantly differ from baseline levels. However, a visual task time exceeding 30 min and how many rest breaks are required to prevent and alleviate visual fatigue and VIMS still need to be explored.

Considering the impact of the movie’s content on participants’ emotional experience, they avoided emotions such as boredom and sleepiness that could affect the experimental results. This study allowed all participants to choose movie content that interested them and all participants chose the drama movie. Depending on the movie’s content, there may be different sound and lighting effects, which may affect the severity of visual fatigue and VIMS; therefore, it is worth further exploring the impact of different movie content on visual fatigue and VIMS in the future.

The current study did not examine the gender effect on visual fatigue and VIMS while using the OHMD. Larese Filon et al. [43] examined 3054 VDT operators by way of follow-ups over 10 years with periodic medical examinations and eye evaluations. The results indicated that visual fatigue was common among VDT operators during follow-up but there was no relationship between visual fatigue and gender. Hsiao et al. [44] reported that the gender effect was not significant on visual fatigue while participants executed a reading task on an OHMD. However, regarding VIMS, Flanagan et al. [45] reported that women report a history of VIMS about twice as frequently as men. Hemmerich et al. [46] indicated that women experiencing severe menstrual pain reported severe VIMS compared to women with low menstrual pain and men. It seems that gender is one factor affecting the occurrence of VIMS; therefore, it is worth further examining the gender effects on work-rest schedules for OHMD visual tasks.

In this study, an OHMD with a fixed head position was used as the experimental equipment. The results may be different when the AR tasks or the presenting modes of the OHMD (such as the world-fixed mode) are changed. Furthermore, the experiments in this study were conducted in a sitting position. It is reasonable to postulate that other visual tasks and use conditions, for example, watching an operation flow video and performing a task simultaneously, may generate different results. Additionally, the current study did not consider the interaction between the context of use and personal factors, such as the nature of the visual task, using the OHMD, and environmental and personal factors (i.e., age, personality, and learning experience), which might also influence the results and warrant further exploration.

## 5. Conclusions

This study aimed to determine work-rest schedules for visual tasks of different lengths by evaluating visual fatigue and VIMS while using an OHMD. The results indicated that after completing the 15 and 30 min visual tasks, the participants experienced visual fatigue and VIMS. For the 15 min visual task followed by a 20 min rest and the 30 min visual task followed by a 25 min rest, the participants’ CFF values, four EEG relative power index values, and SSQ scores were not significantly different from baseline values. Therefore, this study suggests a 20 min rest for visual fatigue and VIMS recovery after a 15 min visual task on an OHMD and a 25 min rest for visual fatigue and VIMS recovery after a 30 min visual task. This study provides a work-rest schedule for OHMD use that can be used as a reference for OHMD user guides to reduce visual fatigue and VIMS and to enhance the safety and health in the workplace.

## Figures and Tables

**Figure 1 ijerph-20-01880-f001:**
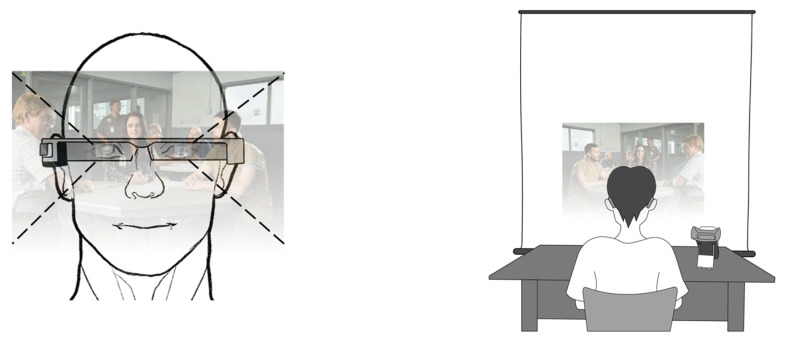
Participants executing a visual task using OHMD in this study.

**Figure 2 ijerph-20-01880-f002:**
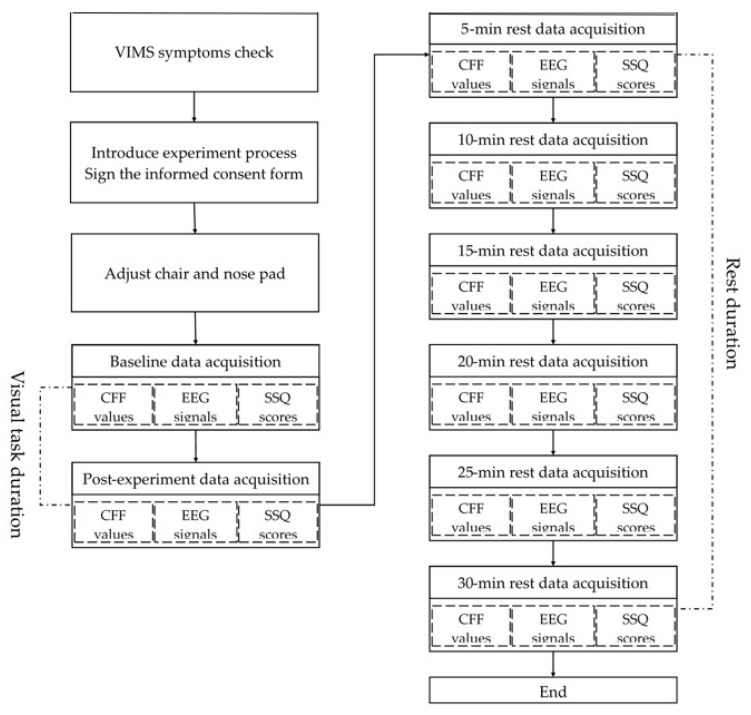
The experimental procedure sequence in this study.

**Figure 3 ijerph-20-01880-f003:**
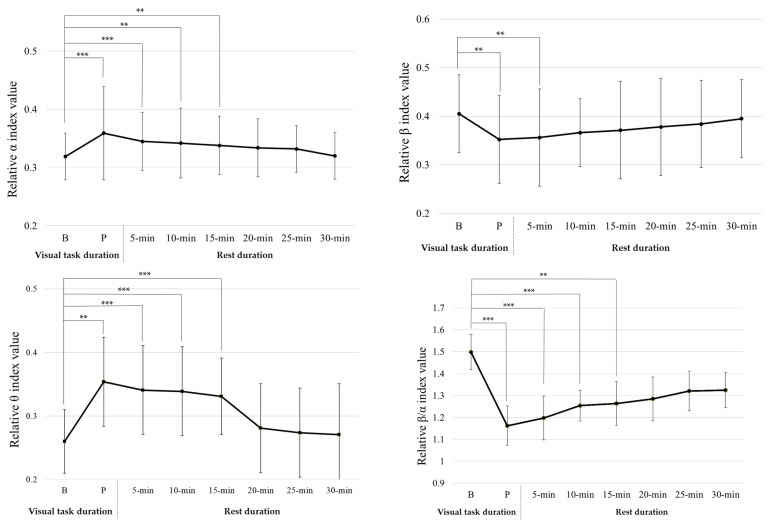
Wilcoxon signed-rank test results for 4 EEG relative power indices for 15 min visual task baseline, post-experiment, and rest time. Note: B: Baseline, *p*: Post-experiment, ** *p* <0.01; *** *p* < 0.001.

**Figure 4 ijerph-20-01880-f004:**
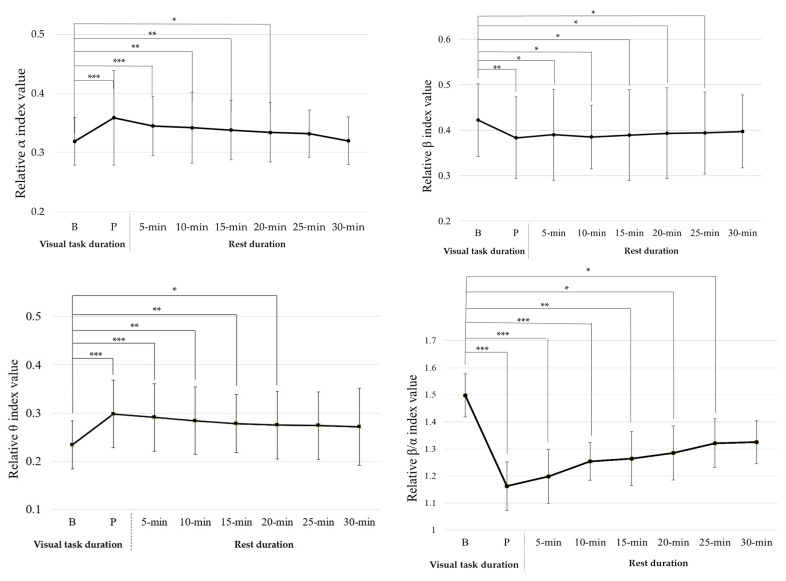
Wilcoxon signed-rank test results for 4 EEG relative power indices for 15 min visual task baseline, post-experiment, and rest time. Note: B: Baseline, P: Post-experiment, * *p* < 0.05; ** *p* <0.01; *** *p* < 0.001.

**Table 1 ijerph-20-01880-t001:** Computation of nausea, oculomotor, disorientation, and total severity score.

SSQ Items	Nausea	Oculomotor	Disorientation
1. General discomfort	v	v	
2. Fatigue		v	
3. Headache		v	
4. Eyestrain		v	
5. Difficulty focusing		v	v
6. Increased salivation	v		
7. Sweating	v		
8. Nausea	v		v
9. Difficulty concentrating	v	v	
10. Fullness of head			v
11. Blurred vision		v	v
12. Dizzy (eyes open)			v
13. Dizzy (eyes closed)			v
14. Vertigo			v
15. Stomach awareness	v		
16. Burping	v		
Total	(1)	(2)	(3)
		SSQ scores	
Nausea		(1) × 9.54	
Oculomotor		(2) × 7.58	
Disorientation		(3) × 13.92	
Total		[(1) + (2) + (3)] × 3.74	

**Table 2 ijerph-20-01880-t002:** Means and standard deviation results for CFF values for each visual task at baseline, post-experiment, and in the rest session.

Source		CFF (Unit: Hz) (Mean ± SD)	
		15 Min Visual Task	30 Min Visual Task
Baseline		35.539 ± 2.964	35.600 ± 3.086
Post-experiment		34.061 ± 2.801	33.583 ± 2.581
Rest duration	5 min	34.712 ± 2.991	34.222 ± 2.796
	10 min	35.334 ± 2.701	34.382 ± 2.841
	15 min	35.282 ± 3.081	35.331 ± 3.201
	20 min	35.424 ± 3.130	35.402 ± 3.241
	25 min	35.491 ± 3.071	35.474 ± 3.014
	30 min	35.522 ± 3.091	35.535 ± 3.122

**Table 3 ijerph-20-01880-t003:** Means and standard deviation results for the different EEG electrode locations.

Source	15 Min Visual Task		30 Min Visual Task	
	Mean ± SD		Mean ± SD	
	O1	O2	O1	O2
α	0.335± 0.065	0.336 ± 0.077	0.323 ± 0.056	0.318 ± 0.058
β	0.377 ± 0.083	0.377 ± 0.082	0.394 ± 0.088	0.389 ± 0.094
θ	0.303 ± 0.071	0.303 ± 0.074	0.268 ± 0.066	0.271 ± 0.071
β/α	1.175 ± 0.355	1.178 ± 0.364	1.291 ± 0.413	1.259 ± 0.407
θ/α	0.922 ± 0.217	0.932 ± 0.244	0.867 ± 0.258	0.864 ± 0.255

**Table 4 ijerph-20-01880-t004:** Means and standard deviation results for SSQ scores for 15 min visual task baseline, post-experiment, and rest time.

15 Min Visual Task		SSQ Scores (Mean ± SD)			
		Nausea	Oculomotor	Disorientation	Total
Baseline		0	3.537 ± 5.535	3.712 ± 9.625	2.618 ± 4.622
Post-experiment		5.088 ± 9.284	17.686 ± 11.623	15.776 ± 21.852	20.944 ± 18.881
Rest duration	5 min	4.452 ± 7.816	11.117 ± 10.854	6.496 ± 10.165	14.711 ± 14.060
	10 min	1.272 ± 3.298	9.601 ± 9.532	4.640 ± 8.442	7.729 ± 8.502
	15 min	1.272 ± 3.298	3.537 ± 5.535	0.928 ± 3.532	1.995 ± 3.064
	20 min	1.272 ± 3.298	3.537 ± 5.535	0.928 ± 3.532	1.995 ± 3.064
	25 min	0	3.032 ± 5.487	0.928 ± 3.532	1.745 ± 3.064
	30 min	0	3.032 ± 5.487	0.928 ± 3.532	1.745 ± 3.064

**Table 5 ijerph-20-01880-t005:** Means and standard deviation results for SSQ scores for 30 min visual task baseline, post-experiment, and rest time.

15 Min Visual Task		SSQ Scores (Mean ± SD)			
		Nausea	Oculomotor	Disorientation	Total
Baseline		0.636 ± 2.420	5.053 ± 7.269	4.640 ± 6.674	3.989 ± 5.283
Post-experiment		6.864 ± 9.201	29.815 ± 17.908	38.048 ± 39.529	28.673 ± 20.539
Rest duration	5.088 ± 5.999	24.761 ± 16.765	33.408 ± 37.919	23.188 ± 19.848	14.711 ± 14.060
	7.314 ± 16.747	17.181 ± 15.792	22.272 ± 29.608	17.204 ± 21.102	7.729 ± 8.502
	4.770 ± 6.007	16.170 ± 17.323	12.992 ± 28.999	13.215 ± 17.672	1.995 ± 3.064
	1.908 ± 5.255	11.623 ± 13.753	6.496 ± 15.394	8.228 ± 11.386	1.995 ± 3.064
	0.636 ± 2.420	6.569 ± 7.377	8.352 ± 19.205	5.735 ± 8.083	1.745 ± 3.064
	0.636 ± 2.420	5.053 ± 6.081	1.856 ± 7.063	3.241 ± 4.578	1.745 ± 3.064

**Table 6 ijerph-20-01880-t006:** Spearman correlation analysis results between the four relative EEG power indices and the CFF value for the visual task duration and rest duration.

	15 Min Visual Task		30 Min Visual Task	
	Visual Task Duration	Rest Duration	Visual Task Duration	Rest Duration
CFF value	−0.302, *p* = 0.019	0.114, *p* = 0.012	−0.362, *p* = 0.254	0.106, *p* = 0.157
α	0.156, *p* = 0.234	−0.156, *p* = 0.068	0.279, *p* = 0.235	−0.191, *p* = 0.074
β	−0.037, *p* = 0.781	0.018, *p* = 0.809	−0.266, *p* = 0.482	0.123, *p* = 0.099
θ	0.227, *p* = 0.081	−0.127, *p* = 0.090	0.214, *p* = 0.821	−0.421, *p* = 0.105
β/α	−0.129, *p* = 0.326	−0.086, *p* = 0.252	−0.356, *p* = 0.121	−0.286, *p* = 0.152

## Data Availability

No new data were created or analyzed in this study. Data sharing is not applicable to this article.

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
