# Peer review of "Determining Work-Rest Schedules for Visual Tasks That Use Optical Head-Mounted Displays Based on Visual Fatigue and Visually Induced Motion Sickness Recovery"

_ijerph, 2023, doi:10.3390/ijerph20031880_

Round 1
Reviewer 1 Report
It is a very interesting study for help relieving visual fatigue.
The topic can be simplified as Work-rest schedule for using optical head mounted displays based on visual fatigue and visually induced motion sickness recovery. The result can be very sensitive based on six levels of resting time. The topic is original because there are other study comparing the performance of using optical head mounted displays. Using brain wave to study visual fatigue and visually induced motion sickness recovery.
Some redundancy such as on page 3 all the participants were required to have normal or correct-to-normal vision. similar statement appeared on page 5 Assessment of visual acuity confirmed that the participants had normal 20/20 vision.
A diagrammatic demonstration of the experimental setup and procedure sequence can be very helpful.
The conclusion reminds me of the whole study statement can be made more clear If the authors can emphazue and divide the manuscript into two major parts Determination of the duration of visual task based on experience of visual fatigue and VIMS. The second part describing when the CFF, EEG, and SSQ were not significant different from the baseline to derive the rest duration for the 15 and 30 minutes of visual task.
The methodology is very detail which was separated and numbered by subsections. Since many measurements were taken simultaneously, it will be helpful to use a graph to present the timing sequence for all of the measurement technuqies. Other parts of the manuscript can become more readable if follow the numbering of the subsections as in the methodology section.
The quality of the tables and figures can be significantly improved.
Reviewer 2 Report
Thanks for the kind review invitation; please see the attached comments for the authors.

Reviewer 3 Report
In this study, a work-rest schedule for OHMD was proposed that can be used as a reference for OHMD user guidelines to reduce visual fatigue and visually induced motion sickness, such as oculomotor symptoms, disorientation and nausea. This study was from a practical point of view, but some questions still need to be addressed further.
1. The introduction part discussed from the indicators used in this study respectively, but this part still lacked descriptions of the background and innovation of this topic.
2.In the material and method part, there are some questions as below:
a) It is noted that the participants used in this study were all men, without taking into account the impact of gender differences. As I know, female and male individuals probably differently respond to stimuli. So, it’s necessary to take gender into careful consideration.
b) The experimental movie content included a documentary, comedy, action, romance, science fiction, or drama, and can be chosen by each participant. But did this design take into account the impact of the movie's content on the emotional experience of the participants? And whether it was balanced among the participants? In the discussion part, the influence of the movie's own characteristics can be discussed.
c) As for the selection of independent variables, two levels of visual task duration (15 minutes and 30 minutes) were used. I hope the authors can give reasons for why choosing these two levels of durations. Maybe the observation of increasing continuous time will make the conclusion more credible. Similarly, the necessity of selecting these three dependent variables also needs to be further explained.
d) Although previous studies had showed that EEG recorded from the optical lobe was suitable for measuring visual reality, but this study only used two electrodes (O1 and O2). In the data analysis part, only one electrode was analyzed. The reliability and effectiveness of the results need to be discussed.
3. In the data description part, some details need to be improved.
a) Table 2, 3, 4, and 5 should be presented in a unified way, generally, it’s not a common way to show the significance in the descriptive statistical tables. In addition, it’s not necessary to present results in both figures and tables simultaneously.
b) SPSS was used for data analysis, and which software version was used for drawing also need to be described. In addition, the way of significance marking in Figure 1 and Figure 2 needs to be standardized.
c) For Figure 1 and Figure 2, the horizontal axes need to be clear. If the horizontal axis represents “time”, the current presentation was not suitable. What’s more, what “B” and “P” represent needs to be clearly stated in the notes, such as "B" for baseline, "P" for post experience.
d) In part 3 “results”, it’s not sufficient to say whether there was a significant difference between the two variables or levels. It’s necessary to explain which one is higher or lower than the other one, and what the results indicated. In addition, the number of decimal places for “p value” should be unified.
4.In the data analysis part and discussion part, the authors mainly discussed CFF values, EEG relative power indicators, and SSQ scores separately. Although it was indicated that Spearman correlation analysis was used to evaluate the relationship between the CFF value and relative EEG power indicators, the relationship between different variables still lacked further analysis.
5.There were also some irregularities in the writing of this manuscript. For example, the p value needs to be italicized. It should be noticed that some sentences missed full stops or repeated punctuation in some sentences.
